# CONVEX IS BACK:
# SOLVING BELIEF MDPS VIA CONVEXITY-INFORMED DEEP REINFORCEMENT LEARNING

## ABSTRACT

We present a novel method for Deep Reinforcement Learning (DRL), incorporating the convex property of the value function over the belief space in Partially Observable Markov Decision Processes (POMDPs). We introduce hard- and soft-enforced convexity as two different approaches, and compare their performance against standard DRL on two well-known POMDP environments, namely the Tiger and FieldVisionRockSample problems. Our findings show that including the convexity feature can substantially increase performance of the agents, as well as increase robustness over the hyperparameter space, especially when testing on out-of-distribution domains.

## 1 INTRODUCTION

Markov Decision Processes (MDPs) have become the standard formalism for solving sequential decision making problems (Van Otterlo & Wiering, 2012). For applications in which perfect observability cannot be assumed, MDPs can be extended to model the probabilistic decision process as a Partially Observable MDP (POMDP), which provides an efficient framework for optimal decision making under uncertainty (Kochenderfer, 2015; Kaelbling et al., 1998; Shani et al., 2013; Braziunas, 2003; Walraven & Spaan, 2019; Oliehoek et al., 2008; Cassandra et al., 1994).

Classical dynamic programming (DP) and reinforcement learning RL are the two established solution methods for MDP and POMDPs. RL approaches are used to overcome the curse of dimensionality of DP methods, and additionally do not require a model of the environment (Kochenderfer, 2015). Neural network (NN) based deep RL (DRL) applied to MDPs has been particularly successful, even in high dimensional problem settings (e.g., Mnih et al., 2013; 2015; Silver et al., 2016; 2018). Solving POMDPs is a more difficult task, but one approach is to directly handle the noisy observation-action history (or a variant thereof) (Koutas et al., 2024; Hettegger et al., 2023; Arcieri et al., 2023). In cases, where no environment model is available, this is the only option. If one has an environment model, then beliefs, i.e., a probability distribution over all system states, can be computed. Using beliefs can be computationally beneficial (Arcieri et al., 2023; Koutas et al., 2024); however, most current approaches do not use DRL with beliefs. In this work, we propose to extend belief-based DRL solutions for POMDPs, by taking into account the theoretical property that the optimal value function is convex over the belief space. We hypothesize that introducing this informative property in the training process enables faster learning, and leads to better performance in out-of-distribution (OOD) domains.

We propose two approaches for convexity-enforcement of the value function in the training process, namely soft-enforced and hard-enforced convexity. The performance of convexity-informed DRL is examined for both convexity approaches applied to the Dueling Q-Network (Wang et al., 2016) value-based architecture. Two benchmarks are used for performance comparison, which are the classic *Tiger* (Kaelbling et al., 1998) and *FieldVisionRockSample* (FVRS) (Ross et al., 2007) problems. We show that when trained with the convexity modification, the involved NNs have stronger generalization performance compared to the standard training schedule and, in some cases, better training performance.

**Related work**
Learning convex functions has a rich history in machine learning. However, the techniques devel-

oped in literature deal with explicitly defined target functions, e.g., including Karush-Kuhn-Tucker conditions of quadratic programs as differentiable layers Amos & Kolter (2017), log-likelihoods in conditional random fields Zheng et al. (2015), or energy functions in structured prediction energy networks Belanger & McCallum (2016). These methods are not applicable to DRL, where the value function is implicitly defined through the Bellman equation, and are thus not similar to our contribution.

Regarding hard-enforced convex NNs for MDP problems, the closest work to this paper is a combination of Amos et al. (Amos et al., 2017) and Sivaprasad et al. (Sivaprasad et al., 2021). Amos et al. (Amos et al., 2017) introduce fully and partially input convex neural networks, where the architectures include required skip connections at every layer. The focus of their work lies more on performing inference on the final convex NN to find the optimal input values. Their reference to the application on DRL focuses merely on perfectly observable and continuous action problems. They represent the negative Q-function with an input convex NN, and subsequently select the optimal action as a convex optimization problem over the NN output. However, they do not consider partially observable environments, as is the focus of this work. Sivaprasad et al. (Sivaprasad et al., 2021) define conditions to achieve input-output convexity with fully connected NNs without the need for skip connections, and convolutional architectures; however, they do not consider DRL implementations.

Regarding soft-enforcement of convexity in NNs for MDPs or POMDPs, we are not aware of any related work.

## 2 BELIEF MDP

Markov Decision Processes (MDP) provide an efficient framework for finding optimal solutions in sequential decision making problems, where the environment $E$ is stochastic, the consequence of the agent's actions is probabilistic, and the state of the environment is known (Puterman, 2014). Partially observable MDPs (POMDPs) provide a natural extension, where the true environment state is not perfectly known; instead the agent receives imperfect observations (Kochenderfer, 2015). The POMDP can be formulated as an MDP by replacing the system states $s_t \in \mathbb{S}$ with the corresponding belief $b(s_t) = p(s_t \mid o_{1:t}, a_{1:t-1})$, where $o_t \in \mathbb{O}$ and $a_t \in \mathbb{A}$ denote the received observation and action performed by the agent, respectively. As new information is available to the agent, the new belief states can be obtained with Bayesian updating (Kochenderfer, 2015; Andriotis & Papakonstantinou, 2019):

$$b(s_{t+1}) = p(s_{t+1} \mid o_{t+1}, a_t, \boldsymbol{b}_t)$$
$$= \frac{O\left(o_{t+1} \mid s_{t+1}, a_t\right)}{p(o_{t+1} \mid \boldsymbol{b}_t, a_t)} \sum_{s_t \in \mathbb{S}} T\left(s_{t+1} \mid s_t, a_t\right) b(s_t), \tag{1}$$

where $O$ and $T$ represent the observation and state transition probabilities, respectively, $p(o_{t+1} \mid \boldsymbol{b}_t, a_t)$ is the normalizing constant, and the belief vector $\boldsymbol{b}_t$ of length $|\mathbb{S}|$ represents the collection of beliefs $b(s_t) \, \forall s \in \mathbb{S}$ (Andriotis & Papakonstantinou, 2019).

Based on the chosen action and the underlying true state of the environment, the agent receives a reward $r_t \in \mathbb{R}$ determined by the reward function $r(s_t, a_t)$, and the expected reward in a certain belief state can be obtained by $r(\boldsymbol{b}_t, a_t) = \sum_{s_t \in S} r(s_t, a_t) b(s_t)$. The decision-making rule, mapping beliefs to actions, is called policy $\pi(\boldsymbol{b}_t)$ [1]. The total expected discounted reward, or *value function*, $V^\pi(\boldsymbol{b}_t)$, starting from $\boldsymbol{b}$ at time $t$ until the horizon $h$ under the policy $\pi$ is defined as (Walraven & Spaan, 2019; Andriotis & Papakonstantinou, 2019):

$$V^\pi(\boldsymbol{b}_t) = \mathbb{E}_{s_k \sim T, a_k \sim \pi, o_k \sim O} \left[ \sum_{k=t}^{h} \gamma^{k-t} r(\boldsymbol{b}_k, a_k) \right], \tag{2}$$

where $\gamma < 1$ defines a discount factor. Similarly to Equation 2, one can define an *action-value* function, $Q^\pi(\boldsymbol{b}_t, a_t)$, which defines the value of taking action $a$ at belief $\boldsymbol{b}$ at time $t$ until the horizon $h$ under the policy $\pi$:

$$Q^\pi(\boldsymbol{b}_t, a_t) = \mathbb{E}_{s_k \sim T, a_k \sim \pi, o_k \sim O} \left[ \sum_{k=t}^{h} \gamma^{k-t} r(\boldsymbol{b}_k, a_k) \,\middle|\, a_t \right]. \tag{3}$$

---

[1] We limit this work to deterministic policies, but an extension to stochastic policies is possible

The relationship between Equations 2 and 3 is $V^\pi(\boldsymbol{b}_t) = Q^\pi(\boldsymbol{b}_t, \pi(\boldsymbol{b}_t))$. Having the Q-values, the agent's policy can be easily extracted by:

$$\pi(\boldsymbol{b}_t) = \arg\max_{a_t \in \mathbb{A}} Q^\pi(\boldsymbol{b}_t, a_t). \tag{4}$$

The goal of the agent is to find the policy which maximizes the expected sum of discounted rewards. This optimal policy $\pi^*$ maximizes the value and action-values at every time $t$:

$$V^*(\boldsymbol{b}_t) := V^\pi(\boldsymbol{b}_t)|_{\pi=\pi^*} = \max_\pi V^\pi(\boldsymbol{b}_t) \tag{5}$$

$$Q^*(\boldsymbol{b}_t, a_t) := Q^\pi(\boldsymbol{b}_t, a_t)|_{\pi=\pi^*} = \max_\pi Q^\pi(\boldsymbol{b}_t, a_t), \tag{6}$$

where $Q^*$ satisfies the recursive Bellman optimality condition (Cassandra et al., 1994):

$$Q^*(\boldsymbol{b}_t, a_t) = r(\boldsymbol{b}_t, a_t) + \gamma \sum_{\boldsymbol{b}_{t+1} \in \mathbb{B}} p(\boldsymbol{b}_{t+1}|\boldsymbol{b}_t, a_t) \max_{a_{t+1} \in \mathbb{A}} Q^*(\boldsymbol{b}_{t+1}, a_{t+1}). \tag{7}$$

The focus of this work revolves around an important property of the optimal POMDP value function, namely that it should be convex over the belief space (Kochenderfer, 2015). The different methods of enforcing this property are discussed in Section 4 and the application of the approaches to the Dueling architecture is discussed in Section 5.1.

## 3 DEEP REINFORCEMENT LEARNING

One can use the recursive formulation in Equation 7 to find the function $Q^*$, from which the optimal policy can then be extracted. Dynamic programming variants, such as, e.g., value or policy iteration, perform the optimization by iterating over all possible combinations of (belief) states, actions and observations (Kochenderfer, 2015). However, due to the super-exponential growth in the value function complexity (Hauskrecht, 2000), this approach is not feasible for larger state and action spaces. By contrast, Neural Networks as universal function approximators (Hanin, 2019) have proven to be effective even in large state and action spaces, which has motivated their use for approximating $V$, $Q$, or $\pi$, in what is known as Deep Reinforcement Learning (DRL) (e.g., Andriotis & Papakonstantinou, 2019; Vinyals et al., 2019).

One of the most prominent architectures for DRL are Deep Q-Networks (DQNs) (Mnih et al., 2013), where the recursive Equation 7 is reformulated into the temporal difference (TD) mean-squared error (MSE) loss function:

$$MSE_t = \frac{1}{n_t} \sum_{i=1}^{n_t} \left| y^{(i)} - \tilde{y}^{(i)} \right|^2 \tag{8}$$

where $y^{(i)}$ denotes an output sample of the NN and $\tilde{}$ always denotes the counterpart of the target NN, whose weights get updated periodically (Mnih et al., 2015):

$$y^{(i)} = Q(\boldsymbol{b}_t^{(i)}, a_t^{(i)} \mid \theta) \tag{9}$$

$$\tilde{y}^{(i)} = r_t^{(i)} + \gamma \max_{a_{t+1}^{(i)} \in \mathbb{A}} \tilde{Q}(\boldsymbol{b}_{t+1}^{(i)}, a_{t+1}^{(i)} \mid \tilde{\theta}). \tag{10}$$

Note that the Q-values in Equations 9 and 10 are DRL approximations of the optimal Q-function in Equation 6, and the TD-MSE in Equation 8 is a sample-based approximation of the Bellman condition in Equation 7. The assumption is that, given enough samples and training, the DRL approximation should converge to the optimal solution.

## 4 CONVEX NEURAL NETWORKS

### 4.1 CONVEXITY CONDITIONS FOR MULTI-DIMENSIONAL FUNCTIONS

A function $f : \mathbb{R}^n \longrightarrow \mathbb{R}$ is convex if its domain is a convex set and for all $\boldsymbol{u}, \boldsymbol{v}$ in its domain, and all $t \in [0, 1]$, we have (Ahmadi & Hall, 2016)

$$f(t\boldsymbol{u} + (1-t)\boldsymbol{v}) \leq tf(\boldsymbol{u}) + (1-t)f(\boldsymbol{v}) \tag{11}$$

If $f$ is differentiable, one can define an alternative condition of convexity, which is equivalent to Equation 11 (Ahmadi & Hall, 2016):

$$f(\boldsymbol{u}) + \nabla_{\boldsymbol{u}} f(\boldsymbol{u})^T (\boldsymbol{v} - \boldsymbol{u}) \leq f(\boldsymbol{v}), \text{ for all } \boldsymbol{u}, \boldsymbol{v} \in \text{dom}(f). \tag{12}$$

If $f$ is twice differentiable, the one can define yet another condition of convexity which is equivalent to Equations 11 and 12 (Ahmadi & Hall, 2016):

$$0 \preceq \boldsymbol{H}(f)(\boldsymbol{u}) = \nabla_{\boldsymbol{u}}^2 f(\boldsymbol{u}), \text{ for all } \boldsymbol{u} \in \text{dom}(f), \tag{13}$$

i.e., the Hessian matrix $\boldsymbol{H}(f)(\boldsymbol{u})$ must be positive semi-definite. Equation 11 defines convexity in terms of the function value at different points, whereas Equations 12 and 13 define convexity via the first and second derivatives.

## 4.2 HARD-ENFORCED CONVEXITY

The first obvious choice to satisfy convexity in the value function is to use an NN architecture which guarantees convexity. Considering a multi-layer perceptron of $k$ layers, where $h_i^{(l)}$ denotes the $i-$th neuron output in the $l-$th layer, then for an input $\boldsymbol{x} \in \mathbb{R}^d$, $h_i^{(l)}$ is defined as (Sivaprasad et al., 2021):

$$h_i^{(l)} = \phi \left( \sum_j W_{i,j}^{(l)} h_j^{l-1} + b_i^{(l)} \right), \tag{14}$$

with weights $W_{i,j}^{(l)}$, bias $b_i^{(l)}$ and activation function $\phi(x)$. Further, $h_j^{(0)} = x_j (j = 1, \ldots, d)$ and $h_j^{(k+1)} = y_j$ ($j^{\text{th}}$ NN output). Only two conditions are needed to ensure convexity of the final output y with respect to the input x, namely (Amos et al., 2017; Sivaprasad et al., 2021):

   i) $0 \leq W_{i,j}^{(2:k+1)}$,

   ii) $\phi$ is convex and a non-decreasing function.

Condition ii) can be achieved by using, e.g., Leaky Rectified Linear Unit (LReLU) (Maas et al., 2013), Parametric ReLU (PReLU) (He et al., 2015) or Exponential LU (ELU) (Clevert et al., 2015) activation functions.

Condition i) needs to be enforced during the training process, e.g., by clipping negative weights to zero, taking absolute of weights, exponentiation of negative weights, or shifting negative weights after each iteration (Sivaprasad et al., 2021). As this enforcement from outside interferes with the weight updates, and hence potentially hinders training, we also investigate the approach of soft-enforced convexity. Note that other, more complex approaches exist to enforce convexity, e.g., by representing weights as separate NN layers with absolute activation functions (Rashid et al., 2020). However, this would change the underlying architecture and complicate comparability between the approaches, which is why other enforcement options are left for future research.

## 4.3 SOFT-ENFORCED CONVEXITY

Soft-enforced convexity mimics the idea of soft-enforced differential equations in Physics-Informed Neural Networks (PINNs) (Raissi et al., 2019). The core idea is to add a second term to the temporal loss function in Equation 8, which penalizes deviations of the target function from one of the convexity criteria outlined in Section 4.1. Suppose we use the MSE loss function also for the second term, then the total MSE loss function is:

$$MSE = MSE_t + c \cdot MSE_c, \tag{15}$$

where the constant $c$ defines the relative weight of the TD- and convexity loss terms.

## 5 METHODOLOGY

### 5.1 CONVEXITY-INFORMED DRL

The NN architecture used throughout this work is the Dueling architecture (Wang et al., 2016) depicted in Figure 1.

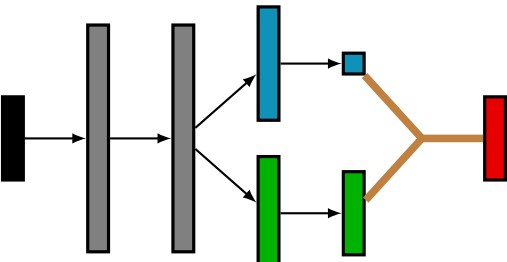

Figure 1: Dueling network architecture, with the belief input (black), dense layers (gray), value stream (cyan), advantage stream (green) and Q-value output (red). Arrows indicate dense weights and the brown lines indicate computation without weights; adapted from (Wang et al., 2016).

**Hard-enforcement** of the value-convexity with respect to the belief input is performed by adjusting the weights according to condition i) in Section 4.2 in the shared layers (gray), as well as the value stream (cyan) in Figure 1. Condition ii) in Section 4.2 is met by an appropriate choice of the activation function, which is kept identical for all layers.

**Soft-enforcement** of the value-convexity is performed by adding a second loss term according to Equation 15. Depending on the choice of the convexity criterion, namely point-based (p) in Equation 11, or gradient-based (g) in Equation 12, $MSE_c$ takes the form of:

$$MSE_c^p = \frac{1}{n_c} \sum_{i=1}^{n_c} \max\left\{0, f(t^{(i)}\boldsymbol{u}^{(i)} + (1 - t^{(i)})\boldsymbol{v}^{(i)}) - t^{(i)}f(\boldsymbol{u}^{(i)}) - (1 - t^{(i)})f(\boldsymbol{v}^{(i)})\right\}^2 \quad (16)$$

$$MSE_c^g = \frac{1}{n_c} \sum_{i=1}^{n_c} \max\left\{0, f(\boldsymbol{u}^{(i)}) + \nabla_{\boldsymbol{u}}f(\boldsymbol{u}^{(i)})^T(\boldsymbol{u}^{(i)} - \boldsymbol{v}^{(i)}) - f(\boldsymbol{v}^{(i)})\right\}^2. \quad (17)$$

The hessian-based condition in Equation 13 cannot be translated into a loss function in a straightforward manner. A matrix $\boldsymbol{M}$ is positive semi-definite (psd) if

$$\boldsymbol{x}^T\boldsymbol{M}\boldsymbol{x} \geq 0 \quad \forall \boldsymbol{x} \in \mathbb{R}^n. \quad (18)$$

For 1D inputs, this is not a problem and the condition reduces to $\frac{d^2}{du^2}f(u) \geq 0$ and the convex loss takes the form of

$$MSE_c^{h,1D} = \frac{1}{n_c} \sum_{i=1}^{n_c} \max\left\{0, -\frac{d^2}{du^{(i)2}}f(u_i)\right\}^2. \quad (19)$$

By contrast, for multidimensional inputs, Equation 18 can be checked in a sample based manner:

$$MSE_c^{h,nD} = \frac{1}{n_c} \sum_{i=1}^{n_c} \frac{1}{n_{psd}} \sum_{j=1}^{n_{psd}} \max\left\{0, -\boldsymbol{x}^{(j)T}\boldsymbol{H}(f)(\boldsymbol{u}^{(i)})\boldsymbol{x}^{(j)}\right\}^2. \quad (20)$$

$\boldsymbol{u}^{(i)}$ and $\boldsymbol{v}^{(i)}$ in Equations 16, 17, 19 and 20 denote points $i = 1, ..., n_c$ sampled from the belief space, for which the respective convexity condition is checked; $\boldsymbol{x}^{(j)}$, $j = 1, ..., n_{psd}$ denote points sampled from $\mathbb{R}^n$ for which the psd condition in Equation 18 checked.

Once the respective soft enforcement method is chosen, belief points $\boldsymbol{b}^{(i)}$ (corresponding to $\boldsymbol{u}^{(i)}$) are sampled from the problem-specific belief space and, together with other inputs, propagated through the network to obtain their values $V(\boldsymbol{b}^{(i)})$ (corresponding to $f(\boldsymbol{u}^{(i)})$). In this work, we employ the Dueling architecture, hence the values can be obtained by propagation through the shared layers (gray) and subsequently through the value stream (cyan) of the network in Figure 1.

### 5.2 NUMERICAL INVESTIGATIONS

With numerical experiments, we test if enforcing convexity in the value function can improve the performance of DRL. Specifically, we test the following two hypotheses:

*H1*: Enforcing convexity enables the DRL agent to learn faster due to restriction of the search space.

*H2*: Convexity-informed DRL performs better in out-of-distribution domains due to improved extrapolation of the value function.

The two hypotheses are tested with experiments on two classic problems, namely the *Tiger* (Kaelbling et al., 1998) and the *FieldVisionRockSample* (FVRS) (Ross et al., 2007) environments. Detailed descriptions of these problems are given in Sections C and D.

To test *H1*, we train the DRL agent with and without enforcing convexity for a fixed number of training steps. We then compare the performance of the individual agents.

To test *H2*, we evaluate the performance of the agents for belief points which are not included in the training distribution. We achieve this by testing on observation functions which differ from the one used to train the agent. For the Tiger problem, we simply change the constant Tiger observation accuracy. For FVRS, the observation accuracy $p_{obs}$ for a rock depends on its Euclidian distance $d$ to the agent. The default (def) observation function is given as $p_{obs}^{def}(d) = 0.5 + 2^{-1-d/d_0}$, where the constant $d_0 = (n-1)\sqrt{2}/4$ is chosen depending on the grid size $n$. To evaluate the agent on OOD data, we define the heaviside (heavi) function $p_{obs}^{heavi}(d) = 1$ for $d \leq d_0$ else 0.5, where $d_0 = 1$. Furthermore, we also define a set of constant (const) observation functions $p_{obs}^{const}(d) = c$, which do not depend on the distance between rock and agent. We use $c \in \{0.5, 0.6, ..., 1.0\}$.

To allow for a fair comparison between the DRL with and without enforcing convexity we fix the hyperparameter optimization procedure beforehand. This prevents bias inflicted during the optimization (e.g., amount of time invested, number of samples, amount of steps). The detailed procedure used is reported in Section E. After the hyperparameter search is conducted, we perform two separate investigations. Firstly, we evaluate the performance of each method over all hyperparameter samples, yielding a rough estimate of the robustness/sensitivity of each method with respect to changing hyperparameters. This approach is not customary in Machine Learning, which is why the corresponding results are only reported in Section B. Secondly, we take the best hyperparameters of each search, and evaluate them over a certain number of runs. The results of this conventional approach are reported in the main text in Section 6.

The final policy of each agent is evaluated with a Monte Carlo (MC) approximation of the expected sum of discounted rewards $\hat{s}_r$. To evaluate the performance of each method, we employ boxplots, comprising the median, interquartile range, and maximum performance, which allows a more complete interpretation of the obtained results.

## 6 RESULTS

### 6.1 COMPUTATION TIME

All computations were performed on an Nvidia Tesla V100 GPU with 16GB RAM. For the Tiger problem, training for 5,000 steps and evaluating the policy with $2 \cdot 10^5$ MC samples took around 5 minutes. For FVRS, training for 50,000 steps and evaluating the policy with $10^4$ MC samples took approximately 60 minutes.

### 6.2 TIGER

We test the Tiger problem for all versions of enforced convexity (hard, point, grad and hess) as well as for the standard approach without enforced convexity.

A visualization of the convexity violation of the standard DRL approach, as well as the corresponding value function correction of our proposed methods is shown in Section B.1.

We perform a hyperparameter search for all convexity methods for various observation accuracies $p_{obs} = \{0.6, 0.8, 0.9, 1.0\}$. The evaluation of our hypotheses over the whole hyperparameter search is outlined in Section B.1.

The test of $H1$ for the best hyperparameters yields no difference between the standard DRL approach and our proposed methods. This is due to the simplicity of the problem, where the majority of agents

finds the optimal policy in the given amount of steps; thus, the mean performance over multiple seeds is close to the optimal performance with only little variation.

To test $H2$ for the best hyperparameters, we train the agents on a specific observation accuracy and cross-evaluate their performance on different observation accuracies. Since again, the majority of the hyperparameters converge to the optimal solution, we perform the cross-evaluation over all optimal agents. We observe that for agents trained on $p_{obs} = \{0.6, 0.8, 0.9\}$ there was no noticeable difference in the cross-evaluation performance between the individual convexity methods. Our explanation for this is that the optimal policy is characterized by two transition points from the action *'listen'* to *'open-left'/'open-right'*. Since these transition points lie very close to each other for $p_{obs} \in (0.5, 1.0)$ the methods do not have to perform a large extrapolation, which leads to almost identical results. For $p_{obs} = 1.0$, however, the agent receives only the belief points $b = 0.5$ and $b = 1.0$ during training; thus, the agent does not know the location of the policy transition. For this case, the results are shown in Figure 2, where the performance on the originally trained observation accuracy is the same for all agents (cf. $H1$), but the performance on $\{0.6, 0.8, 0.9\}$, in a distributional sense, is noticeably worse for the plain DRL approach compared to all convexity-enforced methods. This is reflected by lower medians (blue lines) and/or worse interquartile ranges. This shows that a well-behaved extrapolation of the value function can lead to better performance in out-of-distribution domains. We note that the max over all optimal agents is still the same for all methods. We suspect that this is again due to the simplicity of the Tiger problem.

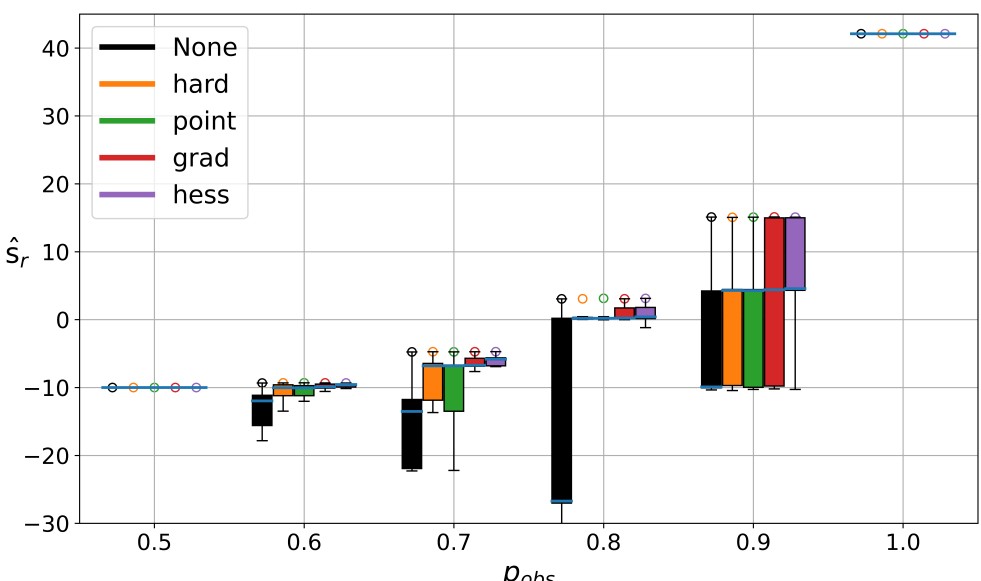

Figure 2: Boxplots (color-coded) over all optimal agents of a hyperparameter search with 200 runs for each convexity method. An optimal agent is one which has reached the optimal policy in the given amount of training steps, and the number of optimal agents was for grad: 178, hard: 68, hess: 69, None: 193, point: 183. The agents have been trained on $p_{obs} = 1.0$ and cross-evaluated on $p_{obs} = \{0.5, 0.6, 0.7, 0.8, 0.9, 1.0\}$ with $10^5$ MC samples. Each boxplot includes the median as a blue horizontal line, interquartile range (IQR) as an opaque colored box, as well as the $1.5 \cdot$IQR distances from the respective quartiles as whiskers; the maximum achieved value is marked with a colored hollow circle, other outliers are not visualized to avoid cluttering.

## 6.3 FVRS

For FVRS, we do not consider the hard-enforced approach, because the value function is convex with regard to the belief inputs but not with regard to the position inputs. However, enforcing convexity of the neural network for only a subset of the inputs is not straightforward, hence we choose to leave this for further research. Moreover, we also do not consider the hessian approach to soft-enforced convexity. Computing second derivatives is simply too time-intensive for larger

problems and one would choose other optimization algorithms (e.g., Newton) over gradient descent if second derivatives were available.

Furthermore, for FVRS we use the LReLU activation functions, as we noticed that during training when the method converges to a stable policy (e.g., always go left), the weights of the hidden layers become high and negative. This ensures that the output is always -1 after passing through multiple ELU layers which yields the same Q-value for every possible combination of inputs. As a result, the method is stuck in this local minimum. To avoid this saturation, we switch to LReLU activation functions for the FVRS problem.

Moreover, we do hyperparameter searches for all convexity methods for the default (def) and heaviside (heavi) observation functions. We report the performances on the originally trained environment as well as the cross-evaluations on other observation functions in the same figures. The results over all samples are reported in Section B.2, whereas the performances of the hyperparameters over 10 runs for are shown in Figures 3 and 4, respectively. When trained on the default setting (Figure 3), both convexity approaches perform better than standard DRL, both on the original and all OOD domains. On the other hand, when trained on the heaviside observation function, the gradient-based approach emerges as the single clear winner over all observation functions.

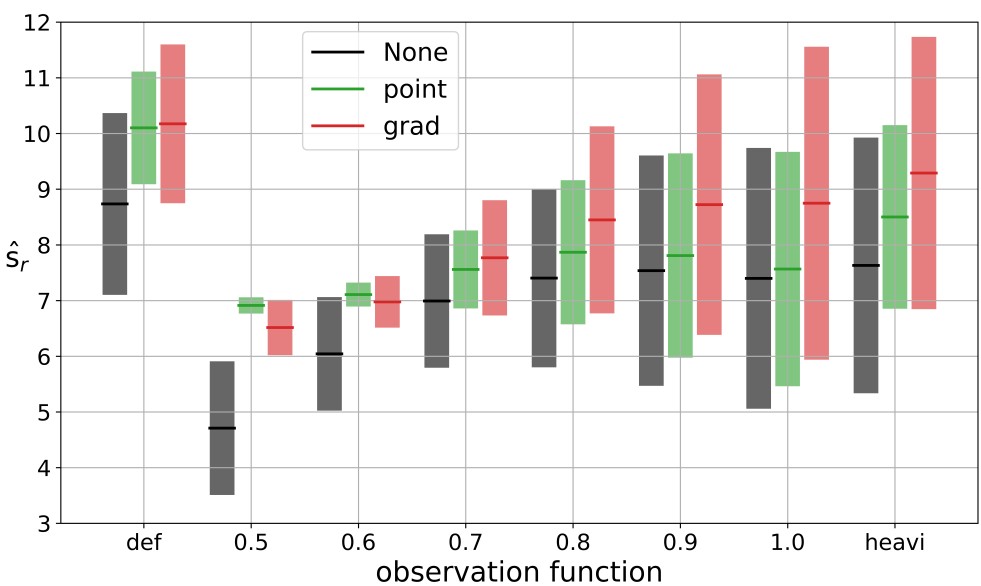

Figure 3: Best agents (color-coded) evaluated for 10 runs for each convexity method. The agents have been trained on the **default** observation function and are cross-evaluated on the heaviside (heavi) and a set of $p_{obs} = \{0.5, 0.6, 0.7, 0.8, 0.9, 1.0\}$ constant observation functions with $10^4$ MC samples. The figure shows the respective reward means (solid horizontal line) as well as $\pm 1$ standard deviation (transparent bars).

## 7 CONCLUSION AND FUTURE WORK

In this work, we propose to extend DRL by enforcing the belief-convexity of the value function in the training process. We have shown that convexity-enforced DRL can yield notable improvements compared to the standard approach, such as better robustness over the hyperparameter space, as well as better mean performance of the best hyperparameters. Our approach performs particularly well when trained on edge case problems ($p_{obs} = 1.0$ for Tiger and $p_{obs} =$ heavi for FVRS) and applying the policy to the standard problem formulation counterpart. This suggests that a well-behaved extrapolation of the value function leads to better policies when encountering OOD-data.

Based on the results in this work, we recommend the usage of the gradient-based enforcement, as it was better or at least equally good compared to the standard and point-based approach in every investigated setting.

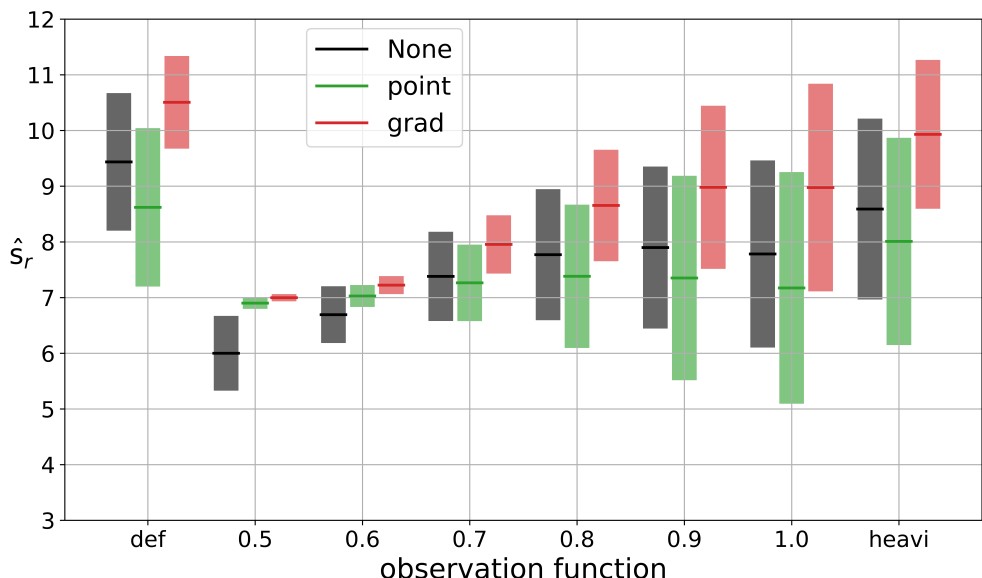

Figure 4: Best agents (color-coded) evaluated for 10 runs for each convexity method. The agents have been trained on the **heaviside** observation function and are cross-evaluated on the default (def) and a set of $p_{obs} = \{0.5, 0.6, 0.7, 0.8, 0.9, 1.0\}$ constant observation functions with $10^4$ MC samples. The figure shows the respective reward means (solid horizontal line) as well as $\pm 1$ standard deviation (transparent bars).

Several new empirical results are presented in this paper, yet there are still numerous open questions to be addressed. In particular, the largest performance gains of the convexity-enforced methods have been observed when training on edge cases, and extrapolating to the standard settings. This strongly suggests that these methods can improve performance when large extrapolations are needed. Hence, we anticipate particularly promising future research directions to be for cases where value extrapolations are required, i.e., in low data regimes, and higher dimensional problems. On the other hand, sampling-based techniques, as our soft-enforced methods, can face scalability challenges. Further investigations of the application to high-dimensional belief spaces are needed to fully grasp the potential of our proposed approaches.

Other potential research directions could be the investigation of convexity-informed DRL using actor-critic architectures, where the target directly is the value function, or the application of these methods to continuous-state POMDPs. Further developments of the convexity injection, e.g., heuristics for choosing an optimal value for $c$ in Equation 15, dynamic $c$ adjustments along the lines of LR-schedules, or including convexity loss every $k$ training steps to speed up the training process, can potentially lead to further improvements and training stability.

## 8    REPRODUCIBILITY STATEMENT

For reproducibility, the source code for the environment, agent, as well as the configuration file containing all relevant parameters is available at the provided website address in the first comment towards the reviewers.

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

# A  CONVEXITY VIOLATION

To check whether the convexity violation of the standard approach is prevalent during training, we plot the value function of 6 example agents trained on the Tiger problem with $p_{obs} = 1.0$ in A.1. Note that not all None-based value functions showed convexity violations, but the majority.

To show that the convexity enforcement approaches proposed in this work mitigate the convexity violations, we also plot the value function of 6 example agents trained on the same setting, but now with gradient-based enforcement. The choice for gradient-based enforcement is arbitrary, all other proposed methods show similar convexity corrections.

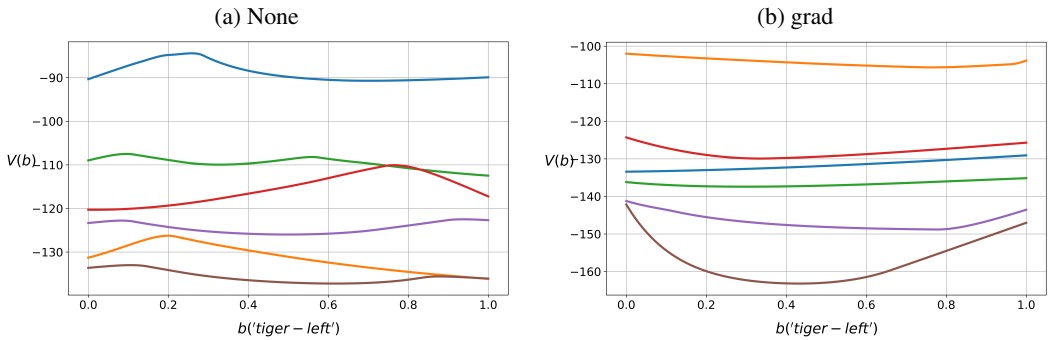

Figure A.1: Tiger value function plot over the belief space for 6 example agents trained without (a) and with gradient-based convexity enforcement (b) on $p_{obs} = 1.0$ .

# B  ROBUSTNESS OVER HYPERPARAMETER SPACE

## B.1  TIGER

To test $H1$ over all hyperparameters, we show their achieved reward distributions for observation accuracies $p_{obs} = \{0.6, 0.8, 0.9, 1.0\}$ in Figure B.1. The figure shows that, in a distributional sense, the performance is significantly worse for the hard-enforced and hessian soft-enforced convexity. Our explanation for this is that training is harder with these convexity enforcements. For the hard enforcement, we suspect that the additional adjustment of the weights after the backpropagation step, which assures that the output of the NN is convex with respect to the input, interferes with the training and hence it is harder to find the optimal policy. For the hessian enforcement, we suspect that calculating third order derivatives is not as stable, which results in slower learning.

## B.2  FVRS

The results for all agents trained on the default, and cross-evaluated on the heaviside and constant observation functions, are shown in Figure B.2. The grad- and point-based methods perform better in all settings compared to the standard approach in terms of their medians and third quartiles ($Q_3$). Note however, that the point method partially shows high variation, which is reflected by low first quartiles ($Q_1$). Overall the grad method shows the best performance based on highest $Q_1 - Q_3$ (highest maximum performance is investigated in Section 6.3).

Even more pronounced is the difference when more extrapolation capacities are needed, i.e., when training on the heaviside and evaluating on default and constant observation functions. The results for this setting are shown in Figure B.3, where the best agents of the grad and point approaches clearly perform better than the None counterpart, both on the original and OOD domains. There does not seem to be a clear difference when comparing the medians of point and None; grad however emerges as the clear best over all observation functions.

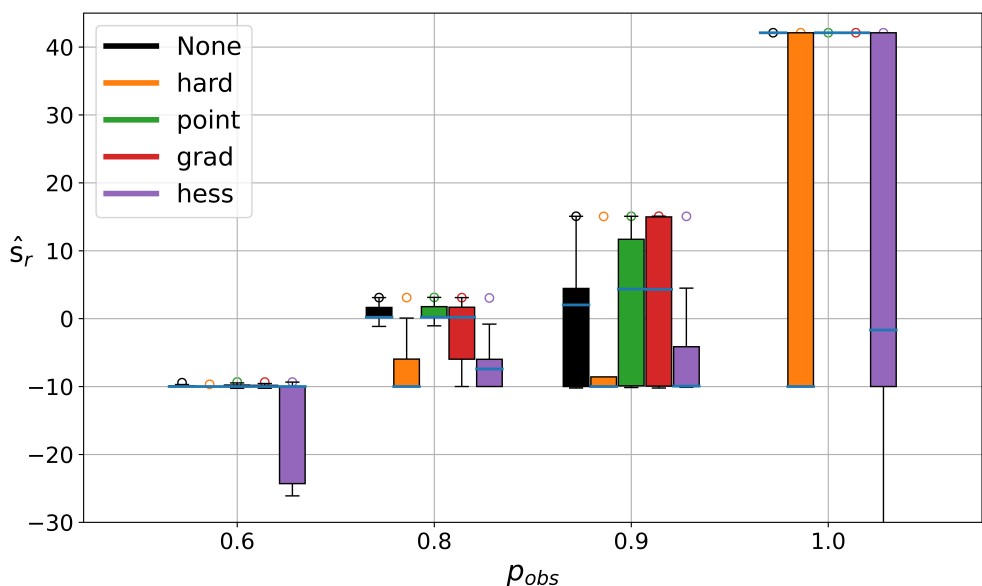

Figure B.1: Boxplots (color-coded) over all agents of a hyperparameter search with 200 runs for each convexity method trained on $p_{obs} = \{0.6, 0.8, 0.9, 1.0\}$, and evaluated with $10^5$ MC samples. Each boxplot includes the median as a blue horizontal line, interquartile range (IQR) as an opaque colored box, as well as the $1.5 \cdot$IQR distances from the respective quartiles as whiskers; the maximum achieved value is marked with a colored hollow circle, other outliers are not visualized to avoid cluttering.

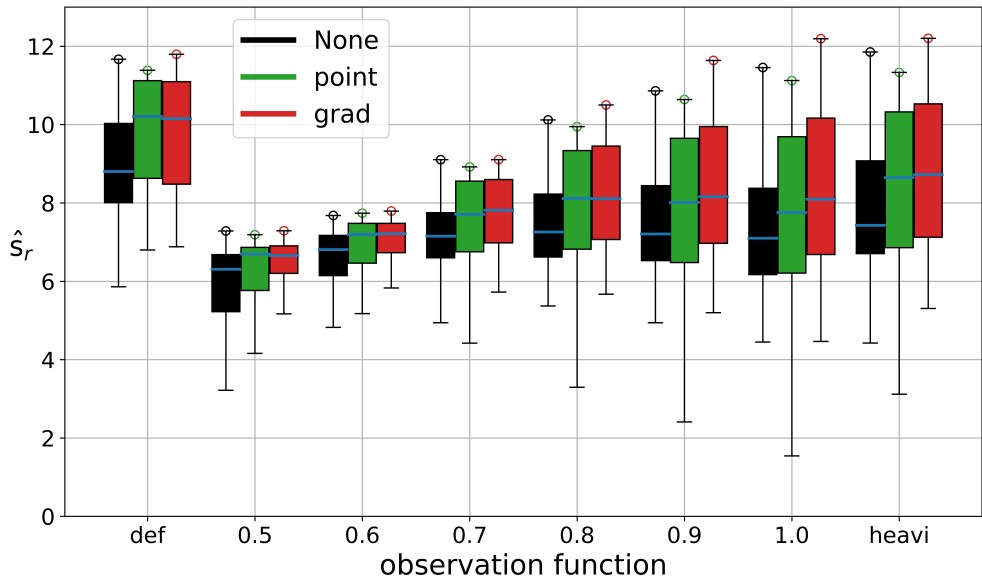

Figure B.2: Boxplots (color-coded) over all agents of a hyperparameter search with 150 runs for each convexity method. The agents have been trained on the **default** (def) observation function and are cross-evaluated on the heaviside (heavi) and a set of $p_{obs} = \{0.5, 0.6, 0.7, 0.8, 0.9, 1.0\}$ constant observation functions with $10^4$ MC samples. Each boxplot includes the median as a blue horizontal line, interquartile range (IQR) as an opaque colored box, as well as the $1.5 \cdot$IQR distances from the respective quartiles as whiskers; the maximum achieved value is marked with a colored hollow circle, other outliers are not visualized to avoid cluttering.

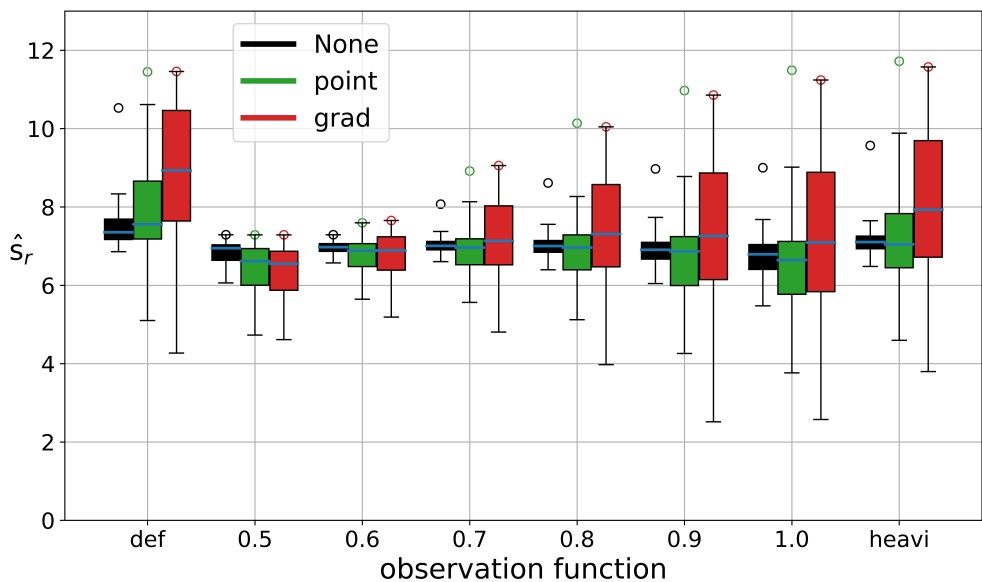

Figure B.3: Boxplots (color-coded) over all agents of a hyperparameter search with 150 runs for each convexity method. The agents have been trained on the **heaviside** (heavi) observation function and are cross-evaluated on the default (def) and a set of $p_{obs} = \{0.5, 0.6, 0.7, 0.8, 0.9, 1.0\}$ constant observation functions with $10^4$ MC samples. Each boxplot includes the median as a blue horizontal line, interquartile range (IQR) as an opaque colored box, as well as the $1.5 \cdot$IQR distances from the respective quartiles as whiskers; the maximum achieved value is marked with a colored hollow circle, other outliers are not visualized to avoid cluttering.

## C  TIGER PROBLEM

### C.1  TIGER DESCRIPTION

The Tiger problem (Kaelbling et al., 1998) consists of an agent standing in front of two doors. Behind one of the doors there is a tiger ($T$), behind the other there is no tiger ($\bar{T}$), and the agent does not know where the tiger is. At each timestep the agent can decide whether he wants to open one of the doors with the actions *'open-right' / 'open-left'* ($O^R$ / $O^L$) or perform a listening ($L$) action. By listening, the agent hears the tiger roaring behind its true location with probability $p_{obs} \in [0.5, 1.0]$.

Opening the door with the tiger behind yields a reward of $r(T) = -100$, opening the other door incurs $r(\bar{T}) = 10$. Listening on the other hand, costs $r(L) = -1$. After opening a door, the environment resets with a new random tiger location.

The environment state of the Tiger problem can be fully described with a scalar value denoting, e.g., the belief of the tiger being behind the left door $b \in [0.0, 1.0]$. In general, the optimal policy depends on the horizon $h$, i.e., the number of times the game is played. For the special cases of $h = 1$ and $h \to \infty$, the optimal policy collapses to a relatively simple one. It consists of listening at every timestep until the belief that the tiger is at a given door falls below a certain optimal belief threshold $b_{opt}$, or the belief surpasses $1 - b_{opt}$. For $h = 1$, the associated optimal belief threshold $b_{opt}^1$ can be obtained by considering only the immediate expected rewards. The agent should open the door if the associated expected reward is higher than the expected reward of listening:

$$\mathbb{E}\left[\mathrm{r}(b, O^{R/L})\right] > \mathbb{E}\left[\mathrm{r}(b, L)\right]$$

$$\iff \quad b(T)r(T) + (1 - b(T))r(\bar{T}) > r(L) \tag{21}$$

$$\iff \quad b(T) < \frac{r(L) - r(\bar{T})}{r(T) - r(\bar{T})} = b_{opt}^1.$$

With the aforementioned rewards, $b_{opt}^1 = 0.1$.

Table C.1: Optimal Q-values for beliefs $b^S = 0.5$, $b^L = 1.0$, $b^R = 0.0$ and actions $L$, $O^L$, $O^R$ for horizon $h \to \infty$ and $r^*_{PO}$ given in Equation 26.

|  | $L$ | $O^L$ | $O^R$ |
|---|---|---|---|
| $b^S$ | $r^*_{PO}$ | $-45 + \gamma r^*_{PO}$ | $-45 + \gamma r^*_{PO}$ |
| $b^L$ | $r^*_{PO}$ | $-100 + \gamma r^*_{PO}$ | $10 + \gamma r^*_{PO}$ |
| $b^R$ | $r^*_{PO}$ | $10 + \gamma r^*_{PO}$ | $-100 + \gamma r^*_{PO}$ |

For $h \to \infty$, the derivation of the thresholds is generally not trivial, as the discounted expected reward of future games has to be considered in addition to the immediate reward, and the threshold depends on the observation accuracy. Thus, we use the *pomdp* package, which is an R implementation of the well-known original *pomdp-solve* developed by Antony Cassandra (Cassandra, 2022; Hahsler & Kamalzadeh, 2021). Instead of calculating the thresholds, we extract the optimal policy graphs together with the associated belief points and optimal expected rewards achieved with discount factor $\gamma = 0.9$ for the investigated observation accuracy cases. Computing the KL-divergence of the optimal policy graphs and the agent policies gives a faster way of determining whether the optimal policy is reached than with extensive MC simulation.

In Sections C.2 and C.3 some special cases resulting from this general optimal policy are outlined, for which the optimal rewards and Q-values can be calculated analytically.

## C.2 UNINFORMATIVE OBSERVATIONS

When $p_{obs} = 0.5$, listening does not yield an improvement of the initial belief (hence, the explored belief space reduces to a single point $b^S = 0.5$). Here, the optimal action is to listen at every timestep and the optimal reward $r^*_{UI}$ and optimal Q-values $Q^*(b, a_i)$ depending on the horizon $h$ and discount factor $\gamma$ are:

$$r^*_{UO} = -\sum_{t=0}^{h} \gamma^t = -\frac{1 - \gamma^{h+1}}{1 - \gamma} \overset{h \to \infty}{=} -\frac{1}{1 - \gamma} \tag{22}$$

$$Q^*(b, L) = r^*_{UO} \tag{23}$$

$$Q^*(b^S, O^L) = 0.5 \cdot 10 + 0.5 \cdot (-100) - \sum_{t=1}^{h} \gamma^t$$

$$= -45 - \gamma \frac{1 - \gamma^h}{1 - \gamma} \overset{h \to \infty}{=} -45 - \frac{\gamma}{1 - \gamma} \tag{24}$$

$$Q^*(b^S, O^R) = Q^*(b^S, O^L) \tag{25}$$

## C.3 PERFECT OBSERVATIONS

When $p_{obs} = 1.0$, listening once directly yields the position of the tiger with certainty. Here, the optimal policy is to listen first and then to open the door opposite to the perceived roar. The optimal reward $r^*_{PO}$ and optimal Q-values for belief states $b^S = 0.5$, $b^L = 1.0$, $b^R = 0.0$ depending on the trajectory until horizon $h$ and discount factor $\gamma$ are given in Equations 26 and C.1

$$r^*_{PO} = -1 + 10\gamma - \cdots + 10\gamma^h = -\sum_{t=0}^{\frac{h-1}{2}} \gamma^{2t} + 10 \sum_{t=0}^{\frac{h-1}{2}} \gamma^{2t+1}$$

$$= -\frac{1 - \gamma^{h+1}}{1 - \gamma^2} + 10 \frac{1 - \gamma^{h+2}}{1 - \gamma^2} \overset{h \to \infty}{=} \frac{10\gamma - 1}{1 - \gamma^2}. \tag{26}$$

# D  FVRS PROBLEM

## D.1  FVRS DESCRIPTION

The FieldVisionRockSample (Smith & Simmons, 2012; Ross et al., 2007) problem is defined by a tuple $(n, k)$, where $n$ defines the width of a square grid and $k$ the number of rocks distributed randomly in the grid. The state of each rock is either good (GR) or bad (BR). The actions available to the agent at each timestep are to move north (MN), south (MS), east (ME), west (MW), or to perform a sampling (C) action. The starting point of the agent is randomly sampled along the west boundary of the grid, and there is an exit zone situated along the east boundary of the grid.

Upon reaching the exit zone the agent receives the reward $r_E = 10$. When the agent is located in the same grid cell as a rock, it can sample it. If the rock is good, the agent receives $r_{GR} = 10$; if it is bad, the agent receives $r_{BR} = -10$. Otherwise, performing the sampling action with no rock present incurs a reward of 0. In addition, we punish the agent if it wants to move out of the grid at the west, north and south boundary. These illegal moves are associated with a reward of $r_{IM} = -10$. Hence overall, in order to maximize the expected sum of discounted rewards, the task of the agent is to sample good rocks and to reach the exit zone in as few steps as possible. Throughout this work we use $(n, k) = (4, 4)$.

## D.2  BELIEF UPDATE

The standard belief update after on observation is given in Equation 1. Firstly, the dependence of $O$ on action $a_t$ can be dropped, since the observations only depend on the state of the rocks and their distance to the robot. Furthermore, an action does not change the state of the rocks [2], thus the transition probabilities reduce to $T(s_{t+1}, s_t) = \delta_{t+1,t}$, where $\delta_{t+1,t}$ denotes the Kronecker delta. Hence, the belief update can be simplified to

$$b(s_{t+1}) \propto O\left(o_{t+1} \mid s_{t+1}\right) b(s_t). \tag{27}$$

## D.3  IGNORING ROCKS POLICY

The first obvious stationary policy is if the agent completely ignores the rocks and collects the exit reward as fast as possible. Thus, the policy consists of choosing the action *"move east"* at every timestep the reward resulting from this policy is

$$r^* = \gamma^{n-1} r_E. \tag{28}$$

The optimal Q-values under the policy of only moving east are then independent of the rock positions as well as the belief about the rock states, i.e., they only depend on the on the position of the agent relative to the exit zone. To simplify, we can formulate an averaged optimal Q-value $\tilde{Q}^*(A) = \mathbb{E}_B\left[Q^*(b, A)\right]$ by averaging over all grid positions.

$$\tilde{Q}^*(ME) = \frac{\gamma^0 r_E + \gamma^1 r_E + \cdots + \gamma^{n-1} r_E}{n} = \frac{r_E}{n} \sum_{t=0}^{n-1} \gamma^t = \frac{1 - \gamma^n}{1 - \gamma} \frac{r_E}{n}. \tag{29}$$

Likewise, for actions $MN$, $MS$, $MW$ we have to take into account the delay of exit reward as well as potential negative rewards incurred due to illegal moves at the borders. Similarly, for action $C$, we have to consider the delay of exit reward as well as the expected reward for performing action $C$ at the agent's location. Since, in our case, the rocks are equally likely to be in a good or a bad state($p(GR) = p(BR)$), the rewards for a good and bad rock have the same absolute value $r_{GR} = -r_{BR}$, and the reward for performing action $C$ when there is no rock is 0, the overall expected collection reward is zero:

$$\begin{aligned}
\mathbb{E}\left[r(C)\right] &= r(C|R)p(R) + r(C|\bar{R})p(\bar{R}) \\
&= \frac{k}{n^2}\left[p(GR)r_{GR} + p(BR)r_{BR}\right] + \frac{n^2 - k}{n^2}r(C|\bar{R}) \\
&= 0.
\end{aligned} \tag{30}$$

---

[2]To be precise, sampling a good rock does change its state to a bad rock; this is however implemented as an additional step after the belief update and hence can be ignored here

Hence, the averaged optimal Q-values for each action are

$$\tilde{Q}^*(MN) = \frac{r_E}{n} \sum_{t=1}^{n} \gamma^t + \frac{r_{IM}}{n} = \gamma \tilde{Q}^*(ME) + \frac{r_{IM}}{n} \tag{31}$$

$$\tilde{Q}^*(MS) = \tilde{Q}^*(MN) \tag{32}$$

$$\tilde{Q}^*(MW) = \frac{r_E}{n} \sum_{t=2}^{n+1} \gamma^t + \frac{r_{IM}}{n} = \gamma^2 \tilde{Q}^*(ME) + \frac{r_{IM}}{n} \tag{33}$$

$$\tilde{Q}^*(C) = \gamma \tilde{Q}^*(ME). \tag{34}$$

### D.4 CONVENIENCE COLLECTION POLICY

Another stable policy with higher reward than completely ignoring the rocks would be to keep moving towards the exit zone at every timestep. However, if a good rock lies incidentally on the path of the agent, it is collected. This policy is always better or equal than ignoring the rocks, when the rewards are selected as

$$\gamma^{n-1} r_E \leq \gamma^n r_E + \gamma^{n-1} r_{GR} \quad \longrightarrow \quad (1-\gamma) r_E \leq r_{GR}, \tag{35}$$

which is guaranteed by the initial setup. To calculate the analytical reward and Q-values resulting from this policy is not straightforward, but can be easily obtained with Monte Carlo simulation.

## E TRAINING SPECIFICATIONS

We first start with a large number of hyperparameters and a broad range (i.e., multiple orders of magnitude) of possible values. For each convexity method, we then sample this space of possible hyperparameters with a fixed number of samples, which we call training runs, and let the agents train for a fixed number of timesteps (no early stopping). The sampling of the hyperparameter space is conducted via via Bayesian hyperparameter tuning (Shukla, 2023), which empirically finds better hyperparameters compared to their more prominent counterparts, namely grid and random search (Bergstra & Bengio, 2012). At the end of each run, we evaluate the policies of the respective agents. In the general case, we estimate the achieved expected sum of discounted rewards with Monte Carlo simulation. For the Tiger problem, an optimal solution is available with classical methods; hence, we can additionally use the Kullback-Leibner divergence to filter for agents which achieved the optimal policy (e.g., in Figure 2).

Since the agents usually find the optimal solution when trained for a sufficiently large number of steps, we heavily restrict the number of available steps for each training run. A few training runs were used for an initial estimate of the speed of convergence and a fraction of that was used for the cutoff. Based on this small pre-analysis, the maximum number of training steps is chosen as 5,000 and 50,000 for the Tiger and FVRS problems, respectively.

For the Tiger problem, we consider the infinite-horizon stationary policy. With a discount factor of $\gamma = 0.9$, the rollout depth $d_T$ was chosen as 150, where the reward's contribution to the sum of discounted rewards is in the order of $10^{-5}$. On the other hand, for the FVRS problem, we search for the policy which maximizes the expected rewards over a game instance, i.e., from the starting point until the robot reaches the end zone. We restrict the maximum rollout depth to $d_{FVRS} = n^2 \cdot k$. Hence, this defines a limit policy of visiting every grid cell and sampling every rock.

## F NEURAL NETWORK SPECIFICATIONS

For this work, we fixed a number of network and optimizer parameters, the specifications can be found in Tables F.1 and F.2. Unless otherwise specified, we use the PyTorch default values. The resulting total number of trainable weights for the Tiger and FVRS networks are given as 174 and 32,106, respectively.

On the other hand, a number of parameters are selected to be optimized. We choose the Bayesian optimization method with an MC approximation of the expected sum of discounted rewards evaluated at the end of each training run as the maximization target. We reduce the learning rate when

Table F.1: Environment-independent fixed network and optimizer parameters

| Parameter | Tiger/FVRS value |
|---|---|
| Architecture type | Dueling (Wang et al., 2016) |
| Target update type | hard |
| Target update period | 3 |
| Batch size | 20 |
| Rollout steps | 25 |
| Discount factor $\gamma$ | 0.9 |
| Optimizer | Adam (Kingma & Ba, 2014) |
| AMSGRAD | Included (Reddi et al., 2019) |
| Minimum learning rate | $10^{-4}$ |
| Initial exploration rate | 0.5 |

Table F.2: Environment-specific fixed network and optimizer parameters

| Parameter | Tiger | FVRS |
|---|---|---|
| # input nodes | 1 | 3k+2 |
| # output nodes | 3 | 5 |
| # FC Layer | 2 | 3 |
| FC layer width | 10 | 100 |
| # value layers | - | 1 |
| Lalue layer width | - | 50 |
| # advantage layers | - | 1 |
| Advantage layer width | - | 50 |
| Activation function | ELU | LeakyReLU |
| Activation func. par. | 1 (scale) | 0.03 (neg. slope) |
| Max epochs | 5,000 | 50,000 |
| Max # frames | 100,000 | 1,000,000 |

the target metric stops improving according to the *ReduceLROnPlateau* scheduler (LRS). For agent exploration we employ the $\epsilon$-greedy scheme. The list of optimizable parameters along with their distributions and environment-specific bounds is given in Table F.3.

Regarding the choice of the weight $c$ in Equation 15, the approach we took was to first train agents without the convexity loss and then to evaluate their convexity losses with respect to a convexity measure of choice. $c$ is then chosen such that the TD-MSE and the average convexity MSE loss are roughly equal. Future work can investigate this topic further, for a more systematic and general handling of this parameter, potentially providing even better performance enhancements.

Table F.3: Optimizable parameters

| Parameter | Distribution | Tiger bounds | FVRS bounds |
|---|---|---|---|
| Initial learning rate | log-uniform | $[e^{-4},\ e^{-1}]$ | $[e^{-7},\ e^{-3}]$ |
| Replay buffer size | int-uniform | $[1,\ 10^5]$ | $[1,\ 10^6]$ |
| # epochs per rollout | int-uniform | $[1,\ 25]$ | $[1,\ 25]$ |
| LRS factor | uniform | $[0.8, 1.0]$ | $[0.8,\ 1.0]$ |
| LRS patience | int-uniform | $[1,\ 10^4]$ | $[1,\ 5 \cdot 10^4]$ |
| # $\epsilon$ steps | int-uniform | $[1,\ 10^4]$ | $[1,\ 10^5]$ |
| Final $\epsilon$ | uniform | $[0.001,\ 0.5]$ | $[0.001,\ 0.5]$ |

