# OpenReview forum: "Convex is back: \\ Solving Belief MDPs via Convexity-Informed Deep Reinforcement Learning"
_ICLR.cc/2025/Conference — Submitted to ICLR 2025_

### Official Review · Reviewer_ZEVp · 2024-10-25

**Soundness:** 4
**Presentation:** 3
**Contribution:** 2
**Rating:** 6
**Confidence:** 4

**Summary:**

This paper proposes to use convexity regularization for value function learning in POMDPs. The authors compared a few different ways to enforce or soft-enforce convexity against a non-regularized value function. The main hypotheses explored are 1) advantage of convexity regularization for learning efficiency and 2) advantage of convexity regularization for OOD generalization. The experiments were performed on the classic tiger problem and field vision rock sample problem. The results show improved median and max performance in OOD settings.

**Strengths:**

I think the method is well-motivated and original to my knowledge. The hypotheses are clear and well-executed at least for the OOD hypothesis.

**Weaknesses:**

* The analysis of learning efficiency is not complete or conclusive. The authors acknowledged that for the tiger problem, it might be too simple to show substantial differences across methods. For the FVRS problem, I don't think the authors studied learning efficiency. Although I think learning efficiency is less crucial for the problem studied.
* The experimental problems/environments might be too simple to understand the potential of the proposed method. I believe both problems can be solved by point-based value iteration, i.e., no stronger function approximation is needed. What would be of interest is to also demonstrate how the proposed method can be applied to for example partially observable continuous state-space problems that are nowadays standard benchmarks for meta RL (e.g., Mujoco problems in [Varibad](https://arxiv.org/abs/1910.08348))

**Questions:**

* For the FVRS results in Fig. 4 and 5, what is the oracle performance, i.e., the POMDP policies that are not OOD for the test environments? It might be useful to show those values to understand the improvement of convexity regularization compared to the oracle.
* How do the authors think this method could be adapted to continuous state POMDPs where the belief states are usually represented as Gaussians with mean and standard deviation as inputs to the value or policy network?

---

### Official Review · Reviewer_g9pG · 2024-11-03

**Soundness:** 2
**Presentation:** 2
**Contribution:** 2
**Rating:** 5
**Confidence:** 3

**Summary:**

The paper introduces hard and soft approaches to enforce convexity in value function approximators for POMDPs. Since value functions for POMDPs are inherently convex in belief states, the authors propose two methods to leverage this property when learning POMDP solutions using DQN. The first "hard enforcement" method involves designing value function approximators as input convex neural networks (ICNNs). The second "soft enforcement" approach adds penalization terms to the DQN loss function to encourage convexity in the value function approximators. Specifically, three types of penalization terms are presented, each corresponding to a condition equivalent to convexity under different smoothness assumptions. These methods are numerically compared with a standard deep RL approach without convexity enforcement. The results demonstrate that (i) the soft enforcement method generally performs better than the hard enforcement method, and (ii) among the soft enforcement methods, gradient-based penalization is the most preferable, outperforming the standard method without convexity enforcement.

**Strengths:**

* The paper proposes exploiting the convexity of the value function in POMDPs when learning solutions using DRL, an aspect that, to the best of my knowledge, has not been emphasized or published in prior work.
* The authors present and empirically analyze various convexity enforcement techniques, offering several hypotheses regarding the strengths and weaknesses of each approach.

**Weaknesses:**

* Although the paper appears to be the first to propose enforcing convexity in value function approximators for POMDPs, the contribution feels incremental and relatively minor. For instance, enforcing convexity through ICNN is simply an integration of ICNN into a dueling DQN, which is quite straightforward and easily conceivable.
* Regarding the soft methods, verifying the convexity of the value function requires sampling multiple belief states. A concern with this sampling approach is its scalability to high-dimensional belief spaces, as the number of samples needed for effective convexity verification would grow exponentially. This issue should be addressed by demonstrating the method on large-scale POMDPs beyond the two problems presented in the paper.

**Questions:**

* It is unclear why using ICNNs performs worse than promoting convexity indirectly through penalization. The paper suggests that adjusting ICNN weights may interfere with training, leading to inferior performance compared to the soft methods. However, this argument is unconvincing, as adjusting the ICNN weights to ensure network convexity - such as projecting negative weights onto the set of nonnegative reals during training - resembles a projected gradient descent approach. A more compelling argument for this phenomenon would be helpful.
* the numerical evaluation could include comparisons with classical POMDP methods, such as PBVI or QMDP, to demonstrate the advantages of using DRL for POMDPs compared to traditional approaches, thereby making the case more convincing for the readers.

---

### Official Review · Reviewer_E6Ux · 2024-11-03

**Soundness:** 3
**Presentation:** 2
**Contribution:** 3
**Rating:** 6
**Confidence:** 3

**Summary:**

This paper introduces a novel approach to solving POMDPs by incorporating the convexity property of the value function over the belief space. The authors propose two methods to enforce convexity in the value function: hard-enforced and soft-enforced convexity. These approaches were evaluated using the Dueling DQN architecture on two POMDP benchmark problems (Tiger and FieldVisionRockSample). Results demonstrate improved median and maximum performance, along with enhanced generalization, particularly in out-of-distribution settings.

**Strengths:**

The paper is generally easy to follow and clearly written, making it accessible to the readers.

The proposed approaches appear to be simple and relatively original.

Introduces soft-enforcement of convexity in NN for MDPs and POMDPs which has better results than hard-enforcement as per their numerical experiments.

**Weaknesses:**

A major concern with the paper is the approach and metrics used to evaluate the performance and conduct numerical results. Specifically, the authors rely on the median and maximum performance metrics across all training runs of a hyperparameter search, without explaining why these choices are appropriate. This approach could obscure a clear comparison between methods and may fail to reflect true performance variability.

To improve rigor and reliability, I suggest an alternative evaluation protocol: first, conduct a hyperparameter search to identify the best-performing parameters for each approach. Then, using these optimized parameters, run each method multiple times (e.g., at least 5 runs) and report the mean and a measure of variability, such as the 95% confidence interval. This approach would provide more robust and interpretable results, offering insights into the stability and consistency of each method. Additionally, including variability measures in the results would allow readers to assess the statistical significance and generalizability of the findings, strengthening the paper’s empirical contributions.

**Questions:**

Why the median and maximum over all training runs of a hyperparameter search was used?

---

### Official Review · Reviewer_Zu7J · 2024-11-08

**Soundness:** 3
**Presentation:** 1
**Contribution:** 2
**Rating:** 5
**Confidence:** 2

**Summary:**

This work proposes methods to exploit the convexity of the belief-based value functions in partially observable Markov decision processes (POMDPs).

In POMDPs, the distribution over the current state - only available when the dynamics are given - is a sufficient statistic for the whole history and sufficient for an optimal policy.
It is well-known that, in POMDPs, the value function - the expected return of any *belief* - is convex with respect to the belief.
Hence, this work argues justifiably, when learning the value function we could try to encode this behavior.
This work proposes several methods of encoding convexity, ranging from hard-coding convexity in a neural network through architecture decisions (positive weights and relu activation functions) to adding loss functions.

The proposed methods for encoding convexity are applied to dual Q-learning and tested on the tiger and rock sample environments, two typical (but small) POMDP problems.


=== post rebuttal ===
I find some of my initial concerns addressed and have raised my score from 3 to 5.

**Strengths:**

I find the direction this work is taking very interesting and potentially promising as, while the convexity of belief-based value functions was known to me, I have not seen work trying to actively exploit this.
Additionally, the paper identifies multiple ways of doing this, rather than just proposing one and run with it.

Overall, even though this strength section is small, I strongly believe that addressing some of the weaknesses described below will result in a good contribution.

**Weaknesses:**

W1: while typically not a huge problem, the quality of the written text needs some extensive polishing.
    There are quite a few harmless mistakes, such as the second sentence in the abstract starting with 'thus' despite the sentence not describing a consequence and that in an MDP the environment is stochastic and the consequence of the agent's actions is probabilistic (is that not the same?).
    Additionally, citations just kind of seem weird (why is a 2019 paper be cited after a statement about beliefs in POMDPs?).
    More importantly, there seem to be quite a few factual mistakes, such as claiming that the convergence of Bellman equation is an assumption (rather than proven).

W2: Unfortunately, while the idea is strong, the empirical evaluation is too weak.
    The number of environments and the reported results is minimal: two problems and only the performance at some point after training (rather than line plots, which could easily shown the sample efficiency that should arise).
    Additionally, other than on experiments where the dynamics were changed during testing, it seems that no significant performance gain was observed.
    Lastly, it is claimed changing the dynamics is a test of generalization and, thus, that the proposed method performing better means it generalizes better.
    However, generalization here should refer to generalizing to *new beliefs* *in the same environment*.
    I am not quite sure what it means that the proposed method works better on a different problem, but I am not quite convinced these experiments demonstrate generalization.

W3: while I appreciate the numerous methods proposed to encourage convexity in the action value function, there must be a literature on learning convex functions.
    Barely mentioning this is problematic: if they were not looked into then this must be done first and, if they were and used in this work, then they should have been more clearly mentioned as previous work rather than claiming their approaches as novel contributions.
    Though I might be wrong and no-one else is interested in learning convex functions.

**Questions:**

Q1: the value function $V$ is known to be piece-wise linear and convex, but I am actually not quite sure what the action value function $Q$ is supposed to be like.
    Could you give some comments on this?

---

### Meta-Review · Area_Chair_XypX · 2024-12-21

**Metareview:**

The paper presents a novel approach to enforcing convexity in value function approximators for POMDPs, showing improved out-of-distribution (OOD) generalization. The method is original, and the empirical results demonstrate its potential benefits, particularly in terms of OOD performance. However, the evaluation was conducted on relatively simple problems (Tiger and FVRS), which may not fully showcase the method's capabilities. Reviewers suggested that more complex and high-dimensional POMDPs, such as continuous-state environments, should be explored to better demonstrate the method's scalability and general applicability. Additionally, the analysis of learning efficiency was limited, and comparisons to oracle performance would provide more context for the reported improvements.

Despite these concerns, the revisions addressed many of the reviewers' questions, improving clarity and convincing them of the method's potential. A few reviewers increased their scores post-rebuttal, appreciating the clearer argument for the method's effectiveness in enhancing OOD generalization. However, more extensive experiments and a discussion of how the method could be applied to continuous-state POMDPs would strengthen the paper’s claims. Overall, the paper is promising but would benefit from further validation on larger-scale problems.

**Additional Comments On Reviewer Discussion:**

During the rebuttal period, the reviewers raised concerns about the limited experimental scope, the simplicity of the benchmark problems, and the incremental nature of the technical contribution. Specific points included the lack of evaluation on larger, high-dimensional POMDPs, the need for comparisons with classical POMDP methods, and skepticism about the scalability of convexity regularization. Additional issues highlighted were the lack of rigorous evaluation metrics and the absence of oracle performance as a baseline in the experimental results.

The authors addressed these concerns by clarifying their choice of benchmarks, emphasizing the novelty of applying convexity regularization in POMDPs, and discussing the general applicability of their methods to other DRL frameworks. They also acknowledged the limitations of their current experiments and outlined potential future work addressing larger-scale problems.

While the authors' responses clarified their contributions and demonstrated their awareness of the limitations, the fundamental concerns about experimental scope and technical impact remain unresolved. The use of toy problems limits the evidence supporting the method’s generality, and the incremental nature of the contribution requires stronger validation to justify acceptance.

In weighing these points, I concluded that the paper represents promising work but is not yet mature enough for publication. The current evidence does not sufficiently support the scalability or robustness of the proposed methods, and rejecting the paper would provide the authors with the opportunity to refine and strengthen their contributions for future submission.

---

### Decision · Program_Chairs · 2025-01-22

Reject